# The Impact of Metabolic Factors and Lipid-Lowering Drugs on Common Bile Duct Stone Recurrence after Endoscopic Sphincterotomy with Following Cholecystectomy

**DOI:** 10.3390/jpm13101490

**Published:** 2023-10-13

**Authors:** Sheng-Fu Wang, Chi-Huan Wu, Kai-Feng Sung, Yung-Kuan Tsou, Cheng-Hui Lin, Chao-Wei Lee, Mu-Hsien Lee, Nai-Jen Liu

**Affiliations:** 1Department of Gastroenterology and Hepatology, Chang-Gung Memorial Hospital, Linkou Medical Center, Taoyuan 333423, Taiwan; b9002076@adm.cgmh.org.tw (C.-H.W.); h12153@adm.cgmh.org.tw (K.-F.S.); flying@cgmh.org.tw (Y.-K.T.); linchehui@adm.cgmh.org.tw (C.-H.L.); r5266@adm.cgmh.org.tw (M.-H.L.); milk1372@cgmh.org.tw (N.-J.L.); 2School of Medicine, College of Medicine, Chang-Gung University, Taoyuan 333323, Taiwan; alanchaoweilee@hotmail.com; 3Division of General Surgery, Department of Surgery, Chang-Gung Memorial Hospital, Linkou Medical Center, Taoyuan 333423, Taiwan

**Keywords:** recurrent common bile duct stones, lipid level, statin, endoscopic retrograde cholangiopancreatography

## Abstract

Background: Recurrent common bile duct stone after endoscopic retrograde cholangiopancreatography is an undesirable problem, even when a following cholecystectomy is carried out. Important factors are the composition and properties of stones; the most significant etiology among these is the lipid level. While numerous studies have established the association between serum lipid levels and gallstones, no study has previously reported on recurrent common bile duct stones after endoscopic sphincterotomy with following cholecystectomy. Materials and methods: We retrospectively collected 2016 patients underwent endoscopic sphincterotomy from 1 January 2015 to 31 December 2017 in Linkou Chang Gung Memorial Hospital. Finally, 303 patients whose serum lipid levels had been checked following a cholecystectomy after ERCP were included for analysis. We evaluated if metabolic factors including body weight, BMI, HbA1C, serum lipid profile, and lipid-lowering drugs may impact the rate of common bile duct stone recurrence. Furthermore, we tried to find if there is any factor that may impact time to recurrence. Results: A serum HDL level ≥ 40 (*p* = 0.000, OR = 0.207, 95% CI = 0.114–0.376) is a protective factor, and a total cholesterol level ≥ 200 (*p* = 0.004, OR = 4.558, 95% CI = 1.625–12.787) is a risk factor of recurrent common bile duct stones after endoscopic sphincterotomy with cholecystectomy. Lipid-lowering drugs, specifically statins, have been shown to reduce the risk of recurrence significantly (*p* = 0.003, OR = 0.297, 95% CI = 0.132–0.665). No factors were found to impact the time to recurrence in this study. Conclusions: The serum lipid level could influence the recurrence of common bile duct stones after endoscopic sphincterotomy followed by cholecystectomy, and it appears that statins can reduce the risk of recurrence.

## 1. Introduction

Recurrent common bile duct (CBD) stones after endoscopic retrograde cholangiopancreatography (ERCP) is an undesirable problem, even when a following cholecystectomy is carried out. Previous studies have reported that the incidence of CBD stone recurrence after endoscopic sphincterotomy (EST) ranges from 4 to 24% [1,2,3]. One retrospective study from Korea found that the recurrence rate of CBD stones was 18.5% after endoscopic treatment with a following cholecystectomy [4]. In patients who had undergone endoscopic papillary balloon dilatation (EPBD), Tsujino et al. found that the rate of CBD stone recurrence was 8.8% [5]; it was 2.4% in patients who had a cholecystectomy after EPBD, and 15.6% in patients who had their gallbladder left in situ with stones, respectively [5]. However, Song et al. found the rate of recurrent CBD stones is comparable between patients who had cholecystectomy and those who did not [2].

In previously reported studies, certain factors have been identified as predictive of recurrent common bile duct (CBD) stones. Song et al. demonstrated that a CBD diameter of 15 mm or larger and the presence of a periampullary diverticulum were potential predictive factors for recurrence following endoscopic extraction of CBD stones [2]. A retrospective case–control study including 457 patients revealed that 9.2% of these patients had recurrent stones, and basket clearance only (OR = 18.25, 95% CI: 1.05–318.35, *p* = 0.046) and older age (OR = 1.02, 95% CI: 1.00–1.05, *p* = 0.023) were risk factors for recurrent cholangitis, according to a multivariate logistic regression analysis [6]. Feng Deng et al. reported that a diameter of bile duct stones ≥ 10 mm was an independent risk factor for recurrent cholangitis after ERCP, and Eun Soo Yoo et al. revealed that the presence of several CBD stones (≥2) was a related factor [4,7].

Another crucial factor contributing to CBD stones is the composition and properties of stones, with lipid levels being the most significant etiology. The biliary lipid profile includes cholesterol, phospholipids, and bile salts. Bile salts are essential for making cholesterol and phospholipids water-soluble in bile juice, forming simple micelles, mixed micelles, unilamellar vesicles, and multilamellar vesicles. While simple micelles and unilamellar vesicles are relatively stable, multilamellar vesicles are prone to initiating the cholesterol nucleation sequence [8,9].

Gallstones may be classified into two categories: first, cholesterol gallstones, that contain more than 50% cholesterol (nearly 75–80% of gallstones), and second, pigment gallstones, comprising less than 30% of cholesterol by weight, which can be furtherly subdivided into black pigment gallstones and brown pigment gallstones [10,11,12,13]. The mechanism of gallbladder stone formation had several factors, including genetic background (the Lith gene, which is involved in the synthesis, transport, and metabolism of cholesterol and bile acids); secretion of cholesterol by the liver; and supersaturated bile, which precipitates cholesterol crystals’ accumulation and formation of sludge in gallbladder. There are some other supplementary factors, comprising mucin and inflammatory changes in the gallbladder, slow intestinal motility, increased intestinal absorption of cholesterol, and altered gut microbiota [14,15,16]. Different pathways of crystallization in phase transition sequences have been documented as functions of the bile salt-to-lecithin ratio, bile salt species, total lipid concentration, and temperature, as well as cholesterol saturation indices [17]. In addition, a study in Denmark confirmed the association between gallstones and insulin resistance, systemic inflammation and genetic obesity, and type 2 diabetes that fasting glucose (OR 1.14, 95% CI [1.05; 1.24]), fasting insulin (OR 1.03, 95% CI [1.01; 1.05]), homeostasis model assessment insulin resistance (OR 1.18, 95% CI [1.02; 1.36]), and metabolic syndrome (OR 1.51, 95% CI [1.16; 1.96]) [18]. Pigment stone disease is predominately present in the rural Orient, in cirrhotic patients, and elderly patients from the United States [19]. These stones are usually associated with bacterial infection and most contain bacilli and cholesterol crystals. The brown pigment stones usually result from bile juice stasis, such as stricture or sphincter Oddi disorder, and infection, which is mainly due to *E. coli*. They can cause the hydrolysis of bilirubin to unconjugated bilirubin; the creation of free bile acids, which can lead to the formation of calcium salts; and the formation of lysolecithin that promotes cholesterol precipitates. However, the formation of black stones may result from altered pH, increased ionized calcium, and unconjugated bilirubin, with a lesser role played by infection. Hemolysis is a known precipitating factor as well [20].

Many studies had proved the association of the serum lipid profile with gallstones’ formation. An abnormal lipid profile was found in more patients with gallstones than without gallstones in a prospective cross-sectional study on 2019 that included 50 patients [21]. One retrospective study including 133 patients found that serum LDL is significantly higher in patients of age > 40 years old who have gallstones [22]. In North India, plasma cholesterol and triglyceride values were significantly higher in male patients with gallstones compared to a control group, but only plasma triglyceride levels were significantly higher than the control group in female patients [23]. Serum lipid levels are significantly abnormal in patients with gallstones, including levels of triglycerides (TG), high-density lipoprotein cholesterol (HDL), low-density lipoprotein cholesterol (LDL), and apolipoprotein B (APOB) [24].

Although cholesterol is a significant component in gallstones, and several previous studies have demonstrated the association between serum lipid levels and gallstone formation, there is no study to our knowledge that explores the relationship with the recurrence of common bile duct stones. Furthermore, there is currently no effective treatment for preventing recurrent common bile duct stones after ERCP. We hypothesize that serum lipid levels could also impact common bile duct stone formation, warranting monitoring and potential treatment. Therefore, this study aims to investigate whether any metabolic factors influence recurrent common bile duct stones after EST, even after cholecystectomy, and assess the therapeutic effects of medications for lowering serum lipid levels.

## 2. Material and Methods

We retrospectively collected 2016 patients who underwent endoscopic sphincterotomy from 2015 to 2017 in Linkou Chang Gung Memorial Hospital. The exclusion criteria were patients who had benign or malignant stricture, bile leakage after surgery, liver transplantation, or post subtotal gastrectomy with B-I anastomosis, B-II anastomosis, or Roux-en-Y anastomosis. Finally, a total of 303 patients with common bile duct stones whose serum lipid profile had been checked and who had undergone cholecystectomy after a first ERCP were enrolled. The study flow chart is shown as Figure 1.

We collected the patients’ characteristics, including age, gender, body weight, body mass index (BMI), metabolic disease with hypertension (HTN), diabetes mellitus (DM), coronary artery disease (CAD) and stroke; fatty liver was diagnosed and classified into mild, moderate, and severe types using an abdominal echo, the duration of follow up from first ERCP to last OPD date, HbA1C level, and a liver function test upon initial diagnosis of cholangitis and when following up. The serum lipid profile that had been checked after the first ERCP, containing LDL, HDL, total cholesterol, and TG, was also collected. Use of lipid-lowering drugs such as statins, fenofibrate, and ezetimibe was recorded if patients were prescribed them after the first ERCP. Patients using ursodeoxycholic acid and aspirin were also documented and for analysis.

The numerical data are presented as means with standard deviation if they conformed to a normal distribution, and median with quartile if they did not conform to a normal distribution. All the categorical data are presented as a frequency and percentage. All the statistical analysis were performed using SAS 9.4 foundation software. An independent *t*-test was used to compare continuous variables that conformed to a normal distribution, and a Mann–Whitney U test was used for variables that did not conform to a normal distribution. A Chi-square test was used for categorical variables between patient groups. Fisher’s exact test was performed when more than 20% of data points presented an expected frequency of <5. *p* values less than 0.05 were considered to indicate clinical significance. We used univariate and multivariate Cox regression analyses to assess the odds ratio of various metabolic risk factors, including body weight, BMI, fatty liver, HbA1C levels and serum lipid levels, for the prediction of common bile duct stone recurrence after endoscopic sphincterotomy and cholecystectomy. In further tests, serum lipid levels were divided into a group above or under the upper normal limit, and were assessed for their impact on common bile duct recurrence. Lipid-lowering medications, ursodeoxycholic acid, and aspirin were also evaluated using univariate and multivariate Cox regression analyses. Moreover, we recorded the duration from first to second ERCP due to CBD stone recurrence, and the duration from cholecystectomy to second ERCP as well. An independent *t*-test was used to analyze if the factors that impact CBD stone recurrence could also impact the duration of recurrence.

## 3. Results

In total, we recruited 303 patients with an indication of common bile duct stones for endoscopic sphincterotomy and then subsequent cholecystectomy. Within an average of 66 months (in Korea), 61 of these patients (20.1%) had recurrent bile duct stones and needed another session of endoscopic retrograde cholangiopancreatography. Male sex was more predominant among these patients (N = 180, 59.4%), without statistical significance between the recurrence and without recurrence groups. The median BMI was 25 kg/m^2^, i.e., overweight, without statistical significance between the two groups. Additionally, there were 213 patients (71.3%) who had at least one metabolic underlying disease such as hypertension, diabetes mellitus, coronary artery disease or stroke. Over half of patients had mild or moderate fatty liver according to liver echo estimations. Liver function tests after initial ERCP presented the following median and quartile values. AST: 101 (41–249), ALT: 139 (46–264), bilirubin: 2.0 (0.9–3.6), r-GT: 212 (107–304), ALK-p: 144 (98–220), and when following up, AST: 25 (18–30), ALT: 23 (15–31). The patients’ characteristics are shown in Table 1.

Metabolic risk factors including body weight, BMI, fatty liver, and serum lipid levels were analyzed for their ability to predict the recurrence of common bile duct stones, as shown in Table 2. The results showed that all the serum lipid levels, including TG (*p* = 0.004, OR = 1.005, 95% CI = 1.002–1.008), LDL (*p* = 0.004, OR = 1.011, 95% CI = 1.003–1.019), HDL (*p* = 0.000, OR = 0.95, 95% CI = 0.926–0.975), and cholesterol (*p* = 0.005, OR = 1.010, 95% CI = 1.003–1.017) had statistical significance according to a univariate Cox regression test, but only HDL (*p* = 0.000, OR = 0.936, 95% CI = 0.908–0.966) had a protective effect according to a multivariate Cox regression test. Moreover, the serum lipid levels were divided into above and below the upper normal limit for further analysis, and all of lipid profiles had statistical significance according to a univariate Cox regression test, as follows: TG ≥ 150 (*p* = 0.000, OR = 3.583, 95% CI = 1.851–6.936), LDL ≥ 130 (*p* = 0.008, OR = 2.28, 95% CI = 1.238–4.199), HDL ≥ 40 (*p* = 0.000, OR = 0.207, 95% CI = 0.114–0.376), cholesterol ≥ 200 (*p* = 0.000, OR = 3.893, 95% CI = 2.122–7.141). However, it seems that only HDL ≥ 40 had a protective effect (*p* = 0.000, OR = 0.199, 95% CI = 0.102–0.388), and cholesterol ≥ 200 (*p* = 0.004, OR = 4.558, 95% CI = 1.625–12.787) increased the risk of recurrence according to a multivariate Cox regression test.

Lipid-lowering drugs including statins, fenofibrate, and ezetimibe were all recorded, and ursodeoxycholic acid and aspirin were documented for analysis, as shown in Table 3. There were 104 patients (34.3%) who used statins for their underlying disease, but only a small number of patients used fenofibrate (N = 9, 3%) and ezetimibe (N = 16, 5.3%). The multivariate Cox regression test revealed that statins could significantly reduce the risk of common bile duct stone recurrence (*p* = 0.003, OR = 0.297, 95% CI = 0.132–0.665), but ezetimibe (*p* = 0.006, OR = 5.618, 95% CI = 1.659–19.025) and ursodeoxycholic acid (*p* = 0.033, OR = 8.050, 95% CI = 1.188–54.573) may increase the risk of recurrence, in contrast.

We further analyzed the association between factors including HDL, cholesterol, statin usage and the duration of recurrence, as shown in Table 4, Table 5 and Table 6. The overall mean duration is 38.9 ± 36.0 months from first to second ERCP, and 36.5 ± 47.9 months from cholecystectomy to second ERCP. However, no factor was shown to influence the time to recurrence, both from first ERCP and cholecystectomy.

## 4. Discussion

Recurrent bile duct stones after ERCP are not uncommon complications, and effective prevention methods are lacking, except for cholecystectomy. Despite the 2017 UK guidelines recommending treatment for choledocholithiasis, which state that all patients with choledocholithiasis or gallstones should undergo cholecystectomy unless there are contraindications for surgery, no definitive methods have been established to prevent the recurrence of bile duct stones after ERCP [25]. However, in our study, 20.1% of patients had bile duct recurrence after ERCP, despite cholecystectomy being carried out. Compared to previous studies, the incidence of common bile duct stone recurrence after endoscopic sphincterotomy also ranged from 4 to 24% [1,2,3]. However, few studies have discussed serum lipid levels and common bile duct stone formation, especially after cholecystectomy, which may influence the intestinal absorption of bile acid and cholesterol. Because common bile duct stone recurrence may be impacted by endoscopic sphincterotomy or endoscopic papillary balloon dilatation [26,27], we only enrolled patients who had undergone sphincterotomy. Most of our patients (N = 213, 71.3%) had at least one metabolic disease such as hypertension, diabetes, coronary artery disease, or stroke; we selected patients from these groups because they regularly have their lipid levels monitored. This also explains why most of our patient population were overweight (BMI = 25.0 (23.0–28.0)) and had a diagnosis of fatty liver according to a liver echo (59.1%).

In our study, we analyzed metabolic factors including BW, BMI, fatty liver and serum lipid levels; the results showed HDL (*p* = 0.000, OR = 0.936, 95% CI = 0.908–0.966) and HDL ≥ 40 (*p* = 0.000, OR = 0.199, 95% CI = 0.102–0.388) could decrease the rate of CBD stone recurrence, and cholesterol ≥ 200 (p = 0.004, OR = 4.558, 95% CI = 1.625–12.787) may increase the risk of recurrence. We reviewed previous studies, which mostly discuss the association between lipid levels and the formation of gallstones. One population-based study in China recruited 2,068,523 patients, and reported that all the serum lipid levels including total cholesterol, TG, HDL cholesterol, LDL cholesterol and apolipoprotein B were significantly different between patients with or without gallstones [24]. Another study focusing on the Hakka population in the Meizhou area of China reported that the serum TG (*p* < 0.001), LDL-cholesterol (*p* = 0.043), total bile acid (*p* < 0.001), and total bilirubin (*p* < 0.001) levels showed significant differences between patients with gallstones and a control group [28]. A prospective study in Taiwan revealed that nonvegetarians with hypercholesterolemia had a 3.8 times higher risk of gallstone formation compared with vegetarians with normal cholesterol (HR = 3.81, 95% CI = 1.61–9.01). Serum TG increased this risk in men (HR = 2.02, 95% CI = 1.03–3.98) and women (HR = 2.43, 95% CI = 1.52–3.90). Increased HDL was associated with a decreased risk in men (HR = 0.22, 95% CI = 0.09–0.52) and women (HR = 0.55, 95% CI = 0.36–0.85) in a UK-based prospective cohort study [29].

The mechanisms of how serum lipid levels impact common bile duct stone recurrence are not clear, and nor is the association of lipid levels with gallstone formation; there have been inconsistent results in previous studies. Based on the existing literature about gallstones, HDL has been found to facilitate the synthesis of hepatic bile acid, and further increases cholesterol solubility in bile [30,31]. Conversely, HDL cholesterol also promotes the transportation of the majority of cholesterol into bile as a vehicle [32]. Human studies and in situ perfused rat liver experiments have reported that plasma HDL, rather than VLDL and LDL, provides unesterified cholesterol to the liver for secreting into bile [32,33,34]. One study implies that free cholesterol in HDL is preferentially absorbed to bile acid rather than secreted into the bile [35]. Additionally, this may explain why our results showed an inverse correlation between HDL level and the CBD stone recurrence rate, even though cholecystectomy had been carried out.

Cholesterol supersaturation of gallbladder bile is an essential cause of gallstone formation [36], although other factors, including the nucleation of cholesterol crystals, the binding together of these crystals with mucin, and the hypomotility of the gallbladder, also play an important role. Several studies have proved that serum total cholesterol has a positive effect on the formation of gallstones [37,38,39], and the correlation between serum and biliary lipid profiles has also been verified [40]. However, some other studies have found that serum total cholesterol has no significant effect on the formation of gallstones [41,42,43]. One previous study compared serum lipid levels before and after cholecystectomy, and reported that total serum cholesterol, HDL, and LDL levels were all significantly decreased after surgery [44]; this may be attributed to accelerated enterohepatic bile flow causing more excretion of bile acids and cholesterol from the liver. In our present study, we found serum total cholesterol had a significant effect only when divided into a value above 200 mg/dL; we think cholecystectomy might increase biliary cholesterol levels, making them high enough to induce common bile duct stone formation, but more experimental research is warranted to further confirm this. There was no difference in the biliary lipid composition, fasting gall bladder volume, and CCK levels of hypertriglyceridemia patients, but they did have reduced gall bladder emptying times compared with patients with normal TG, according to a previous study [45]; this may explain why TG had no role in the recurrence of common bile duct stones after cholecystectomy.

Although cholesterol is an important and predominant component of gallstones, they are a little different from CBD stones. The presence of CBD stones can be categorized as either primary or secondary, depending on whether they form concomitantly with existing gallstones. Primary CBD stone are also composed predominantly of bilirubin and associated with bile stasis and bacterial infection, while secondary CBD stones are from existing gallstones that pass through the cystic duct; thus their components may be similar to those of gallstones [46]. Although pathogenesis of primary and secondary CBD stones is different and brown stones are the main component of primary CBD stones, cholesterol stones still made up 31.1% of primary CBD stones in patients in a previous study [47]. In our results, we found an association between serum lipid levels and CBD stone recurrence after cholecystectomy, although it is unfortunate that we cannot recognize the type of CBD stones during ERCP; thus, we believe that cholesterol may play an important role in CBD stone formation in these patients, most often accompanied by metabolic disorder. In the future, we aim to determine whether the impact of serum lipid levels differs between brown stones and cholesterol stones. Additionally, we intend to identify the specific population that would benefit from modulating serum lipid levels.

Statins, as lipid-lowering drugs, are prescribed mostly for treating hypercholesterolemia and cardiovascular disease, and act as competitive inhibitors of 3-hydroxy-3 methylglutaryl-coenzyme A (HMG-CoA) reductase, the rate-limiting enzyme that regulates the synthesis of cholesterol in the mevalonate pathway [48]. Statins have been proven to decrease the risk of gallstones’ formation via various mechanisms, including inhibition of hepatic cholesterol biosynthesis by the HMGCR enzyme and activation of the PPAR-γ receptor (a nuclear hormone receptor) [49]. A meta-analysis including 622868 participants from six studies revealed that statin usage could decrease the risk of gallstone disease compared with non-use, especially after cholecystectomy due to gallstone disease [50]. A nested case–control study in Korea also found that statin use for over 545 days was associated with a lower incidence of gallstones (OR = 0.91, 95% CI = 0.86–0.96) [51]. A number of studies conducted in Europe have also shown consistent results [52,53,54]. In our study, 104 patients (34.3%) were prescribed statins, and this significantly reduced the risk of CBD stone recurrence after cholecystectomy (*p* = 0.003, OR = 0.297, 95% CI = 0.132–0.665), which may be due to lower biliary cholesterol levels that result from taking a statin.

Surprisingly, ursodeoxycholic acid and ezetimibe may increase the risk of common bile duct stone recurrence, based on our results. However, the number of patients using these agents were very small; in particular, only five patients used ursodeoxycholic acid. Ezetimibe may selectively inhibit intestinal cholesterol absorption by suppressing the uptake of biliary and dietary cholesterol across the brush border membrane of the small bowels, via the NPC1L1 pathway [55]. However, intestinal NPC1L1 facilitates the absorption of cholesterol, whereas hepatic NPC1L1 increases cholesterol uptake by hepatocytes and decreases bile cholesterol supersaturation [56]. Ezetimibe has been proven an efficacious treatment for hypercholesterolemia, whether as monotherapy or in combination with statins. Moreover, it has also been found capable of preventing gallstones by effectively reducing intestinal cholesterol absorption and biliary cholesterol secretion; it was found to protect gallbladder motility function by desaturating bile in mice. There is also a human study designed to collect bile content by stimulating gallbladder emptying using an intraduodenal infusion of 30 mL of an 80 g/L amino acid solution; this study found that ezetimibe could reduce biliary cholesterol saturation [57]. One animal model study also found ezetimibe caused dose-dependent reductions in biliary cholesterol levels, and prevented gallstone formation [58]; however, mice’s lack of hepatic NPC1L1 should be take into consideration. In contrast, a randomized control trial recruited 40 patients and classified patients into simvastatin, ezetimibe, combined treatment (simvastatin + ezetimibe), or placebo groups, then monitored their plasma and biliary lipid levels after cholecystectomy; this revealed thatbiliary cholesterol was relatively increased when only using ezetimibe, although not to a statistically significant extent [59]. Most existing evidence is based on animal models, and there is a lack of human clinical trials regarding preventing gallstones, not to mention recurrent common bile duct stones after cholecystectomy; thus, we think more significant studies are needed in the future.

There are some limitations to our study: first, this is a retrospective and single-center study. Second, although we collected the complete lipid profile after the first ERCP had been carried out, lipid levels may fluctuate and are easily impacted by diet changes and the effects of medication. Third, we did not consider the drug dosage and the duration of medical treatment, which may have impacted the results. Fourth, we cannot identify the component of the gallstones that was not revealed in the pathologic report after cholecystectomy. Fifth, there is selection bias; this is due to the fact that most patients enrolled had metabolic disease, with serum lipid levels that needed to be monitored.

In summary, this study marks the first attempt to explore the connection between metabolic factors and recurrent common bile duct stones, to the best of our knowledge. Our findings indicate that HDL and total cholesterol levels may influence the recurrence rate of common bile duct stones following ERCP, especially after cholecystectomy. Therefore, we recommend routine monitoring of serum lipid levels after ERCP procedures. Additionally, statin therapy appears promising in reducing the recurrence of common bile duct stones in patients with hyperlipidemia. We anticipate that this study will significantly impact current clinical practices, providing a more effective approach to preventing recurrent common bile duct stones beyond the usage of ursodeoxycholic acid. However, it is worth noting that while ezetimibe holds promise for hyperlipidemia and potentially the prevention of gallstone formation, our study suggests an increased risk of common bile duct stone recurrence with its use. Further research with larger sample sizes is necessary to validate these findings.

## Figures and Tables

**Figure 1 jpm-13-01490-f001:**
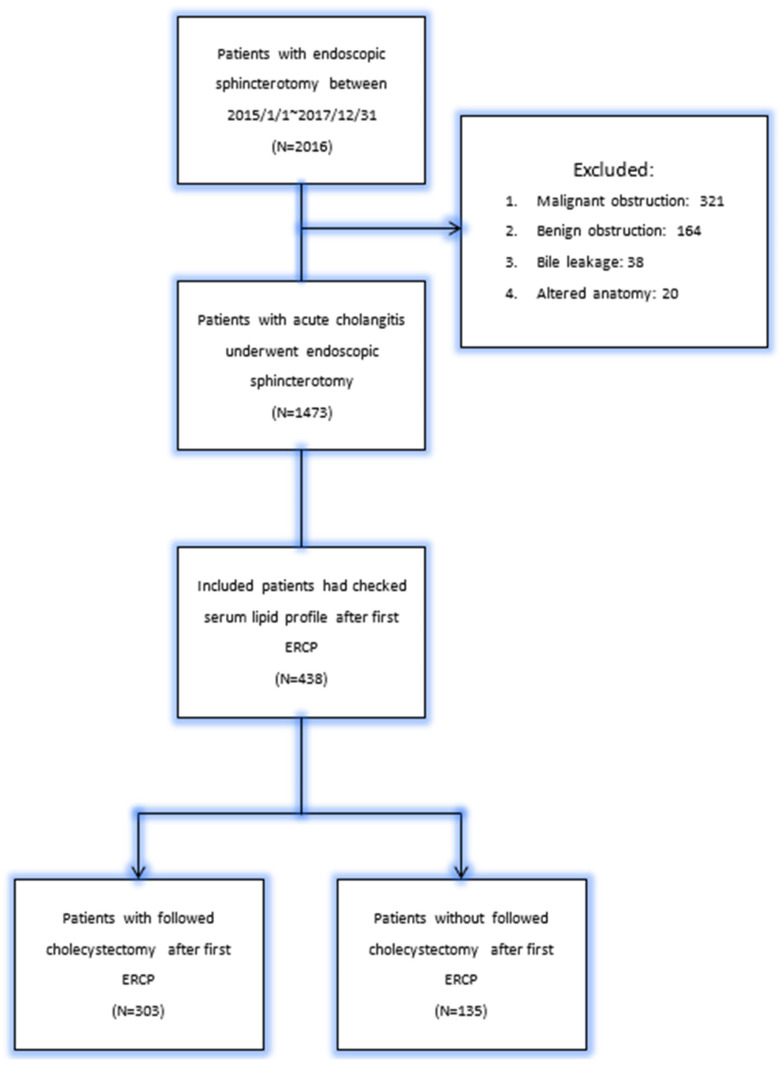
Flow diagram of patient selection.

**Table 1 jpm-13-01490-t001:** Patient characteristics.

	Overall (N = 303)	with Recurrence (N = 61)	without Recurrence (N = 242)	*p* Value
Age (years)	65 (55–74)	68 (60–78)	63 (54–73)	0.031 *
Gender				0.607
Male	180 (59.4%)	38 (62.3%)	142 (58.7%)	
Female	123 (40.6%)	23 (37.7%)	100 (41.3%)	
BW (kg)	67 (58–75)	64 (59–70)	68 (57–75)	0.138
BMI (kg/m^2^)	25.0 (23.0–28.0)	25.0 (23.1–27.3)	25.0 (23.0–28.0)	0.894
Underlying disease				
HTN	168 (55.4%)	38 (62.3%)	130 (53.7%)	0.228
DM	104 (34.4%)	23 (37.7%)	81 (33.5%)	0.534
CAD	27 (8.9%)	4 (6.6%)	23 (9.5%)	0.470
Stroke	14 (4.6%)	4 (6.6%)	10 (4.1%)	0.420
Fatty liver				0.692
No	124 (40.9%)	23 (37.7%)	101 (41.7%)	
Mild	93 (30.7%)	24 (39.3%)	69 (28.5%)	
Moderate	79 (26.1%)	14 (23.0%)	65 (26.9%)	
Severe	7 (2.3%)	0 (0.0%)	7 (2.9%)	
HbA1C (%)	6.1 (5.7–6.5)	6.4 (5.9–6.7)	6.0 (5.7–6.5)	0.013 *
Liver function test upon diagnosis of cholangitis				
AST (U/L)	101 (41–249)	62 (31–189)	107.5 (46.0–274.0)	0.002 *
ALT (U/L)	139 (46–264)	85 (38–188)	155 (67–292)	0.005 *
Total bilirubin (mg/dL)	2.0 (0.9–3.6)	1.6 (0.7–2.6)	2.0 (0.9–4.0)	0.021 *
r-GT (U/L)	212 (107–304)	181 (105–228)	255 (108–308)	0.011
ALK-p (U/L)	144 (98–220)	128 (90–195)	146 (101–229)	0.103
Liver function test upon follow-up				
AST (U/L)	25 (18–30)	25 (20–32)	24 (18–30)	0.095
ALT (U/L)	23 (15–31)	24 (17–34)	23 (15–31)	0.298
Follow-up time (months)	66.0 (51.1–82.0)	79.9 (58.6–91.6)	65.0 (50.1–81.3)	0.007 *

Abbreviations: BW: body weight; BMI: body mass index; HTN: hypertension; DM: diabetes mellitus; CAD: coronary artery disease; AST: aspartate aminotransferase; ALT: alanine aminotransferase; r-GT: r-glutamyl transferase; ALK-p: alkaline phosphatase-p. *: *p* < 0.05 as statistical significance.

**Table 2 jpm-13-01490-t002:** Metabolic risk factors for recurrent cholangitis.

	Overall (N = 303)	with Recurrence (N = 61)	without Recurrence (N = 242)	Univariate		Multivariate	
				OR (95% CI)	*p*-Value	OR (95% CI)	*p*-Value
BW (kg)	67 (58–75)	64 (59–70)	68 (57–75)		0.166		
BMI (kg/m^2^)	25.0 (23.0–28.0)	25.0 (23.1–27.3)	25.0 (23.0–28.0)		0.087		
Fatty liver					0.692		
No	124 (40.9%)	23 (37.7%)	101 (41.7%)				
Mild	93 (30.7%)	24 (39.3%)	69 (28.5%)				
Moderate	79 (26.1%)	14 (23.0%)	65 (26.9%)				
Severe	7 (2.3%)	0 (0.0%)	7 (2.9%)				
Lipid profile							
TG (mg/dL)	117 (81–165)	161 (114–222)	108 (79–149)	1.005 (1.002–1.008)	0.004 *		0.574
TG ≥ 150	49 (16.2%)	20 (32.8%)	29 (12.0%)	3.583 (1.851–6.936)	0.000 *		0.084
TG < 150	254 (83.8%)	41 (67.2%)	213 (88.0%)				
LDL (mg/dL)	97 (76–125)	107 (81–139)	92 (74–120)	1.011 (1.003–1.019)	0.004 *		0.756
LDL ≥ 130	70 (23.1%)	22 (36.1%)	48 (19.8%)	2.28 (1.238–4.199)	0.008 *		0.874
LDL < 130	233 (76.9%)	39 (63.9%)	194 (80.2%)				
HDL (mg/dL)	45 (39–53)	39 (32–49)	46 (40–54)	0.95 (0.926–0.975)	0.000 *	0.936 (0.908–0.966)	0.000 *
HDL ≥ 40	226 (74.6%)	29 (47.5%)	197 (81.4%)	0.207 (0.114–0.376)	0.000 *	0.199 (0.102–0.388)	0.000 *
HDL < 40	77 (25.4%)	32 (52.5%)	45 (18.6%)				
Cholesterol (mg/dL)	173.0 ± 42.1	186.9 ± 58.4	169.5 ± 36.2	1.010 (1.003–1.017)	0.005 *		0.146
Cholesterol ≥ 200	68 (22.4%)	27 (44.3%)	41 (16.9%)	3.893 (2.122–7.141)	0.000 *	4.558 (1.625–12.787)	0.004 *
Cholesterol < 200	235 (77.6%)	34 (55.7%)	201 (83.1%)				
HbA1C (%)	6.1 (5.7–6.5)	6.4 (5.9–6.7)	6.0 (5.7–6.5)		0.120		0.454
HbA1C ≥ 6.5	91 (30%)	25 (41.0%)	66 (27.3%)	1.852 (1.033–3.319)	0.038 *		0.695
HbA1C < 6.5	212 (70%)	36 (59.0%)	176 (72.7%)				

Abbreviations: BW: body weight; BMI: body mass index; TG: triglyceride; LDL: low-density lipoprotein; HDL: high-density lipoprotein. *: *p* < 0.05 as statistical significance.

**Table 3 jpm-13-01490-t003:** Protective factors of medication.

	Overall (N = 303)	with Recurrence (N = 61)	without Recurrence (N = 242)	Univariate		Multivariate	
				OR (95% CI)	*p*-Value	OR (95% CI)	*p*-Value
Statin	104 (34.3%)	12 (19.7%)	92 (38.0%)	0.399 (0.202–0.790)	0.008 *	0.297 (0.132–0.665)	0.003 *
Fenofibrate	9 (3.0%)	2 (3.3%)	7 (2.9%)		0.874		0.810
Ezetimibe	16 (5.3%)	6 (9.8%)	10 (4.1%)		0.084	5.618 (1.659–19.025)	0.006 *
Ursodeoxycholic acid	5 (1.7%)	3 (4.9%)	2 (0.8%)	6.207 (1.014–38.004)	0.048 *	8.050 (1.188–54.573)	0.033 *
Aspirin	53 (17.5%)	7 (11.5%)	46 (19.0%)		0.171		0.603

*: *p* < 0.05 as statistical significance.

**Table 4 jpm-13-01490-t004:** Time to recurrence with HDL.

	Overall (N = 61)	HDL ≥ 40 (N = 29)	HDL < 40 (N = 32)	*p*-Value
From first ERCP to recurrence (months)	38.9 ± 36.0	39.3 ± 36.0	38.5 ± 36.6	0.935
From cholecystectomy to recurrence (months)	36.5 ± 47.9	37.7 ± 55.9	35.4 ± 40.2	0.856

**Table 5 jpm-13-01490-t005:** Time to recurrence with cholesterol.

	Overall (N = 61)	Cholesterol ≥ 200 (N = 27)	Cholesterol < 200 (N = 34)	*p*-Value
From first ERCP to recurrence (months)	38.9 ± 36.0	36.3 ± 36.2	40.9 ± 36.3	0.622
From cholecystectomy to recurrence (months)	36.5 ± 47.9	34.8 ± 51.6	37.9 ± 45.5	0.803

**Table 6 jpm-13-01490-t006:** Time to recurrence with statin use.

	Overall (N = 61)	with Statin (N = 12)	without Statin (N = 49)	*p*-Value
From first ERCP to recurrence (months)	38.9 ± 36.0	38.6 ± 32.9	39.0 ± 37.0	0.978
From cholecystectomy to recurrence (months)	36.5 ± 47.9	25.8 ± 39.4	39.1 ± 49.8	0.390

## Data Availability

Data is unavailable due to privacy or ethical restrictions.

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
