# Peer review of "The Impact of Metabolic Factors and Lipid-Lowering Drugs on Common Bile Duct Stone Recurrence after Endoscopic Sphincterotomy with Following Cholecystectomy"

_jpm, 2023, doi:10.3390/jpm13101490_

Round 1

Reviewer 1 Report

The study looked at patents who had undergone endoscopic sphincterotomy and cholecystectomy and found that high LDL levels were protective against recurrent CBD stobes., while high levels of total cholesterol were a high risk. Additionally, the use of statin drugs was found to reduce the risk of recurrence.

The study has several limitations, including its retrospective and single-center design, which may limit the generalizability of the findings. The fluctuating nature of lipid levels, potential impact of dietary changes or medication effects, and lack of consideration for drug dosage and treatment duration are additional limitations. The study also does not provide information about the composition of the gallstones, and the presence of selection bias is acknowledged. Overall, while the study provides valuable insights, these limitations should be considered when interpreting its findings.

Author Response

Thanks for the important comments, this is our pilot study to prove the association between metabolic factors and recurrent common bile duct stone formation that is not mentioned in previous study although some limitations existed and that were all listed in the part of discussion in our manuscript for readers. In the future, we plan to conduct a multi-center study and solve these limitations as possible. Your acceptance will be the best encouragement to help us keep working hard on this important topic. 

Reviewer 2 Report

The article is well written and covers an important topic. The material, method and exclusions of the work as well as the results are reliable and presented in a transparent way. The discussion is written correctly. As a limitation of the method, I suggest taking into account the possibility of leaving small deposits during ERCP, which may be a limitation not only of this work, but of many works in the world literature. I congratulate the authors on their work.

Author Response

Thanks for the important comments, bile juice and stone extracted when ERCP for analysis is very important evidence as your suggestion. We will take into account this idea in the future when conducting the prospective study and the component of bile and common bile duct stone must play an important role in this topic.

Reviewer 3 Report

Thank you for the opportunity to review this important manuscript. Here are my comments and suggestions.

Excellent idea for the study and excellent results.

In the part of the abstract ‘’ Results: Serum HDL level≥40 (p=0.000, OR=0.207, 95% CI=0.114-0.376) is a protective factor and total cholesterol level≥200 (p=0.004, OR=4.558, 95%’’ the space should be inserted before ‘’ ≥’’ and units for HDL and cholesterol inserted.

Please check English in the following ‘’ Of lipid lowering drugs, statin could reduce the risk of recurrence (p=0.003, OR=0.297, 95% CI=0.132-0.665). Not found any factor may impact the time to recurrent in this study’’. And also in this sentence ‘’ Serum lipid level could impact recurrent common bile duct stone after endoscopic sphincterotomy with followed cholecystectomy and statin seems can reduce the risk of recurrence.’’

Please check is ‘’following cholecystectomy’’ is a correct term.

Change ‘’ One retrospective study conducted in Korea’’ to ‘’ One retrospective study from Korea’’

Check English in ‘’ was 18.5% after endoscopic treatment although cholecystectomy done’’

In the introduction, where the pathophysiology of gallstone formation is explained, please include the pathophysiology of black and brown pigment gallstone formation, which is also very important and pathophysiologically distinctive of cholesterol gallstone formation. This is especially important because this study analyzes common bile duct stone recurrence.

The authors claim that ‘’ Moreover, there’s no effective treatment for preventing recurrent CBD stone after ERCP to date’’. Again, the pathophysiology of brown pigment stones should be discussed.

This study would be outstanding if the gallbladder stone composition after cholecystectomy was analyzed. This could lead us to the underlying cause. At least the composition of the recurrent common bile duct stones.

Minor changes are needed. I pointed to several places in the manuscript so the authors can make changes.

Author Response

Thanks for these important comments and our reply as followed.

  1. We’ve inserted the space before ‘’ ≥’’ and units for HDL and cholesterol inserted in the abstract.
  2. We’ve corrected the sentences in abstract as Page 1, line 30~34.
  3. We’ve corrected the “following cholecystectomy” to “followed cholecystectomy” in manuscript as Page 1, line 40.
  4. We’ve corrected ‘’ One retrospective study conducted in Korea’’ to ‘’ One retrospective study from Korea’’ as Page 1, line 42.
  5. We’ve corrected ‘’ 18.5% after endoscopic treatment although cholecystectomy done’’ to ‘’ 18.5% after endoscopic treatment with followed cholecystectomy done’ as page 2, line 1~2.
  6. The pigment stone especially brown stone is really an important issue that may result in recurrent cholangitis as well. We’ve added the pathophysiology of pigment stone disease in the introduction as Page 2, line 42~51.
  7. Because the importance of brown stone as main component of CBD stone and the different pathophysiology between gallstone and CBD stone, we’ve added the discussion as Page 9, line 49~53 and Page 10, line 1~11.
  8. The composition of common bile duct (CBD) stones or gallstones provides valuable information for our analysis. Unfortunately, we were unable to access these data based on the previous medication records. However, it is crucial to document this information in future prospective studies to enhance the comprehensiveness of our research.

Reviewer 4 Report

First of all, I appreciate the opportunity to review this interesting article on the recurrence of gallstones in patients undergoing ERCP and laparoscopic sphingerectomy with subsequent cholecystectomy, especially the relationship with serum lipid levels and anti-lipemic agents.

In general, the structure of the article is adequate, an introductory theoretical framework, methodology, results and discussion are clearly described. And it seems to me that in its current state, it could be considered for publication, I only recommend reviewing the writing in English again, especially in the abstract and the introduction. Below, I leave some minor observations that I think would be appropriate to take into account:

-While it is true that the limitations of this study are clearly mentioned, the title of the article seems slightly biased to me. Due to the selection bias of the population, and, especially, the lack of certainty of the association with the event being studied (due to the many factors due to which the lipid profile could be elevated in patients who presented recurrence of stones), I consider it important to restructure the title to emphasize its descriptive nature, rather than its impact.

-I suggest separating the discussion into discussion and conclusions, to emphasize the authors' own conclusions (which could be what was mentioned in the last paragraph).

I only recommend reviewing the writing in English again, especially in the abstract and the introduction. Only minor corrections need to be made.

Author Response

Thanks for the comments, we’ve reviewed the writing in English of our manuscript and do our best to refine the English writing.

Reviewer 5 Report

Extensive editing and grammar checks of the manuscript are necessary.

There are  two different font types and sizes used in the body of the manuscript.

This is only a single center study. Collaborating with one or more centers will boost the results and drive more reliable conclusions. 

Extensive editing and grammar checks of the manuscript are necessary.

Author Response

Thanks for the important comments.

  1. We’ve checked the English editing and grammar of our manuscript as good as possible and find a native English speaker to help us to refine the English.
  2. We’ve corrected the font type and size of our manuscript formally, very thanks for your careful inspection that make our work better.
  3. This is our pilot study, and we are willing to conduct a multi-center study in the future to strengthen the validity of our findings, thanks for your suggestion that make us to keep working hard on this important topic.

Round 2

Reviewer 5 Report

 No more comments. The flow of the manuscript  is ok now.